# Strain-induced creation and switching of anion vacancy layers in perovskite oxynitrides

Takafumi Yamamoto [1,2], Akira Chikamatsu [3], Shunsaku Kitagawa[4], Nana Izumo[1], Shunsuke Yamashita [5], Hiroshi Takatsu[1], Masayuki Ochi[6], Takahiro Maruyama[3], Morito Namba[1], Wenhao Sun [7], Takahide Nakashima[1], Fumitaka Takeiri [1], Kotaro Fujii [8], Masatomo Yashima [8], Yuki Sugisawa[9], Masahito Sano[3], Yasushi Hirose [3], Daiichiro Sekiba[9], Craig M. Brown [10], Takashi Honda [11], Kazutaka Ikeda [11], Toshiya Otomo [11], Kazuhiko Kuroki[6], Kenji Ishida[4], Takao Mori [5], Koji Kimoto [5], Tetsuya Hasegawa[3] & Hiroshi Kageyama [1,12,13]✉

Perovskite oxides can host various anion-vacancy orders, which greatly change their properties, but the order pattern is still difficult to manipulate. Separately, lattice strain between thin film oxides and a substrate induces improved functions and novel states of matter, while little attention has been paid to changes in chemical composition. Here we combine these two aspects to achieve strain-induced creation and switching of anion-vacancy patterns in perovskite films. Epitaxial $SrVO_3$ films are topochemically converted to anion-deficient oxynitrides by ammonia treatment, where the direction or periodicity of defect planes is altered depending on the substrate employed, unlike the known change in crystal orientation. First-principles calculations verified its biaxial strain effect. Like oxide heterostructures, the oxynitride has a superlattice of insulating and metallic blocks. Given the abundance of perovskite families, this study provides new opportunities to design superlattices by chemically modifying simple perovskite oxides with tunable anion-vacancy patterns through epitaxial lattice strain.

[1] Department of Energy and Hydrocarbon Chemistry, Graduate school of Engineering, Graduate School of Engineering, Nishikyo-ku, Kyoto 615-8510, Japan. [2] Laboratory for Materials and Structures, Tokyo Institute of Technology, Yokohama 226-8503, Japan. [3] Department of Chemistry, The University of Tokyo, Tokyo 113-0033, Japan. [4] Department of Physics, Graduate School of Science, Kyoto University, Kyoto 606-8502, Japan. [5] National Institute for Materials Science, Ibaraki 305-0044, Japan. [6] Department of Physics, Osaka University, Toyonaka, Osaka 560-0043, Japan. [7] Department of Materials Science and Engineering, University of Michigan, Ann Arbor, MI 48109, USA. [8] Department of Chemistry, School of Science, Tokyo Institute of Technology, Tokyo 152-8551, Japan. [9] Tandem Accelerator Complex, University of Tsukuba, Ibaraki 305-8577, Japan. [10] Center for Neutron Research, National Institute of Standards and Technology, Gaithersburg, MD 20899, USA. [11] Institute of Materials Structure Science, High Energy Accelerator Research Organization (KEK), Tsukuba, Ibaraki 305-0801, Japan. [12] CREST, Japan Science and Technology Agency (JST), Kawaguchi, Saitama 332-0012, Japan. [13] Institute for Integrated Cell-Material Sciences (iCeMS), Kyoto University, Sakyo-ku, Kyoto 606-8501, Japan. ✉email: kage@scl.kyoto-u.ac.jp

n oxides, the introduction of anion vacancies brings about a diversity of chemical and physical properties; the most widely being studied in perovskite oxides[1–3]. If the anion-vacancy concentration ($\delta$ in $ABO_{3-\delta}$) is a rational fraction of the oxygen stoichiometry in the unit cell, the vacancies tend to aggregate to form linear or planar defects. For example, fast oxygen diffusion in $BaInO_{2.5}$ ($\delta = 1/2$) results from oxygen-vacancy chains[4]. In $SrFeO_2$ ($\delta = 1$), $(001)_p$ defect planes of the original perovskite cell allow metal-metal bonding between square-planar Fe(II) centers, leading to a half-metallic state under pressure[5]. The superconducting $T_c$ in cuprates strongly depends on the number of $CuO_2$ sheets made by introducing $(001)_p$ defect planes[6]. Compounds having such anion-defect chains or planes can be synthesized through a variety of approaches such as cationic substitution[7], topochemical and electrochemical reactions[8,9], and under appropriate conditions (temperature[10], gaseous atmosphere[11], or pressure[12]).

Concurrently, advances in the materials science of perovskite-based systems has been amplified with the development of single-crystal (epitaxial) thin films. In particular, lattice strain through a mismatch between the underlying substrate and the deposited film is a key parameter that has been extensively studied. Strain-driven phenomena has led to charge/orbital order in $La_{1-x}Sr_xMnO_3$[13], improved ferroelectricity in $BaTiO_3$[14], multiferroicity in $EuTiO_3$[15], and superconductivity in $La_{1.9}Sr_{0.1}CuO_4$ (ref. [16]). Furthermore, a ferroelectric response in tensile-strained $SrTaO_2N$ films is ascribed to a change in local coordination geometry[17]. The lattice mismatch also allows for the introduction of random oxygen vacancies[18–21]. Controlling vacancy ordering of perovskite oxides have also been reported[22–27], but these efforts are limited to controlling crystallographic orientation of the deposited films such as $Ca_2Fe_2O_5$ brownmillerite.

In this study, we show a low-temperature reaction of $SrVO_3$ (600 °C in $NH_3$ gas) topochemically transforming to $SrVO_{2.2}N_{0.6}$ ($\delta = 0.2$) with regular $(111)_p$ anion-vacancy planes. This is already a surprising observation as such anion-vacancy order has never been seen in oxynitrides. The crystal and electronic structure of $SrVO_{2.2}N_{0.6}$ is mainly two-dimensional, with conducting octahedral layers separated by insulating tetrahedral layers. Most surprisingly, the same ammonia treatment of an epitaxial $SrVO_3$ film on different substrates can change the periodicity of the $(111)_p$ plane, or can even alter the direction of anion-vacancy plane to $(112)_p$, which is distinct from the previous efforts[22–27] where the crystallographic orientation of the film is altered depending on the substrates. This observation suggests that lattice strain can be used to induce and manipulate the anion-vacancy planes and provide a controllable parameter for the development of exotic structural and electronic states in perovskite films.

## Results and discussion

**A $(111)_p$ superlattice in nitridized $SrVO_3$.** Oxynitrides (oxide-nitrides) exhibit attractive properties including visible-light responsive photocatalysis[28], but the highly reducing atmosphere of high-temperature reaction with ammonia (ammonolysis) often makes it difficult to obtain the desired structures[29]. Anion-vacancy order, which is common in oxides, has not been reported in oxynitrides. Recently, low-temperature ammonolysis (≤500 °C) using oxyhydrides has been proven to be a useful approach to access highly nitridized $BaTiO_3$ through topochemical H/N exchange (e.g., $BaTiO_{2.4}N_{0.4}$ from $BaTiO_{2.4}H_{0.6}$)[30]. Subsequently, ammonolysis at moderate temperatures (~800 °C) has been shown to promote topochemical O/N exchange ($EuTiO_2N$ from $EuTiO_{2.8}H_{0.2}$)[31]. The present study extends these approaches to a vanadium oxyhydride[32]. When reacted at low temperature, 300 °C, in $NH_3$, $SrVO_2H$ is topochemically converted into a cubic perovskite $SrVO_{2.7}N_{0.2}$ (Supplementary Note 1, Supplementary Figs. 1 and 2 and Supplementary Table 1). In contrast to the titanium case, even the pristine oxide $SrVO_3$ can be partially nitridized at 500 °C resulting in $SrVO_{2.85}N_{0.1}$, while the nitrogen content is lower than that obtained from $SrVO_2H$ at 300 °C (Supplementary Note 1 and Supplementary Fig. 1). Both oxynitrides carry tetravalent vanadium ions.

With a moderate increase in the ammonolysis temperature to 600 °C, the ex situ X-ray diffraction (XRD) patterns of $SrVO_3$ and $SrVO_2H$ become drastically different (Fig. 1a and Supplementary Fig. 1), although the sample color remains black. Since the XRD profiles are very similar, the data for the $SrVO_3$-derived sample will be shown in the main body of the manuscript. The resulting structure has a rhombohedral cell ($a = 5.51$ Å and $c = 34.3$ Å), similar to that of 15R-type perovskite $SrCrO_{2.8}$ ($Sr_5Cr_5O_{14}$) with oxygen-vacancy order along $(111)_p$[33]. The topochemical nature of

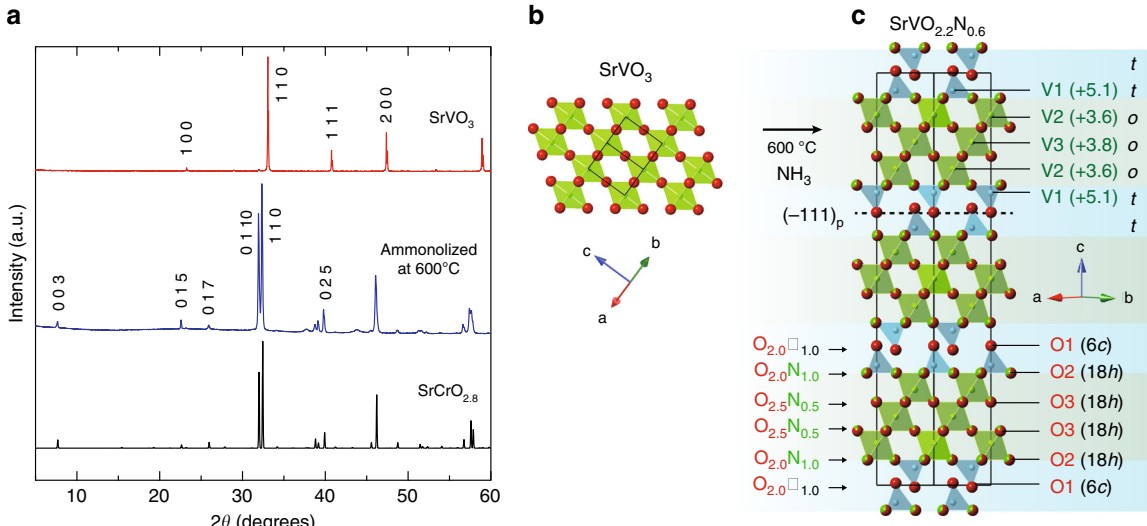

**Fig. 1 Topochemical ammonolysis of perovskite vanadate $SrVO_3$. a** XRD patterns of precursor $SrVO_3$ with the cubic perovskite structure (red, top) and the one ammonolized at 600 °C (blue, middle). A simulation pattern of 15R-$SrCrO_{2.8}$ ($a = 5.51$ Å and $c = 34.5$ Å)[33] is shown for comparison (black, bottom). **b, c** The structural change from $SrVO_3$ (**b**) to 15R-$SrVO_{2.2}N_{0.6}$ ($Sr_5V_5O_{11}N_3$) with anion vacancies in every fifth $(111)_p$ layer, leading to the $(ooott)_3$ stacking sequence (**c**). Sr atoms are omitted for simplicity. Black lines in each structure represent the unit cell.

the reaction is verified by ammonolysis reactions of $Sr_2V_2O_7$ above 600 °C, which yielded $Sr_3V_2O_8$ as the main product (Supplementary Fig. 1c). The 15R phase decomposes under $NH_3$ above 800 °C.

Rietveld refinement to the synchrotron XRD data was carried out assuming the 15R-structure[33] for $SrV(O,N)_{2.8}$ within the $R\bar{3}m$ space group (Supplementary Note 1, Supplementary Fig. 2 and Supplementary Table 2). A subsequent neutron refinement enabled us to determine the anion distribution between N and O, where the site fractional occupancies ($g$) was constrained to $g_O + g_N = 1$. This analysis concluded that the isolated tetrahedral ($6c$) site is fully occupied by oxygen ($g_{O1} = 1$), while nitrogen atoms are partially populated at the $18h$ sites ($g_{N2} = 0.325(8) \approx 1/3$ and $g_{N3} = 0.173(5) \approx 1/6$), resulting in the average chemical formula $SrVO_{2.203(8)}N_{0.597(8)}$. X-ray and neutron refinement for the $SrVO_2H$-derived sample gave similar results, with a composition of $SrVO_{2.22(5)}N_{0.58(5)}$ (Supplementary Note 1, Supplementary Fig. 2 and Supplementary Table 3). Combustion analysis validated the total nitrogen content (Supplementary Note 1). We thus concluded that the composition is $SrVO_{2.2}N_{0.6}$ ($Sr_5V_5O_{11}N_3$).

15R-$SrVO_{2.2}N_{0.6}$ contains one-third of the anion vacancies in every fifth $SrO_3$ $(111)_p$ planes and the residual oxide anions in the $SrO_2$ plane are re-organized to form a double-tetrahedral layer (Fig. 1c). Bond valence sum calculation for the tetrahedral (V1) site gave a value close to 5 (+5.1), while those for the octahedral (V2, V3) sites are +3.6 and +3.8 (Supplementary Table 4). The average valence of +4.2 well agrees with the value expected from the chemical composition. This valence assignment is fully supported by $^{51}V$-NMR as shown later. The transformation of $SrCrO_3$ to 15R-$SrCrO_{2.8}$ can be understood by the crystal field stabilization of $Cr^{4+}$ ($d^2$) in tetrahedral coordination[33]. Since the tetrahedral vanadium in our case is exclusively pentavalent[34–36], the tetrahedral coordination preference should also be the origin of this phase stabilization where $SrV^{4+}O_3$ (or $SrV^{4+}O_{3-2x/3}N_x$) is oxidized by aliovalent $N^{3-}/O^{2-}$ exchange. Although diffusion of nitride anions requires larger activation energy relative to oxide anion[37], perovskite-based compounds with $AO_2$ layers have shown rapid oxide ion conduction[38–40]. In this regard, the diffusion of the nitride ions might be promoted by "in situ" formation of $SrO_2$ layers during the ammonolysis reaction at 600 °C.

## Two-dimensional electronic states

The cubic perovskite $SrVO_3$ has a single $V^{4+}$ site and is characterized as a Pauli paramagnetic metal with a 3D Fermi surface[41]. However, the insertion of periodic anion-vacancy layers in $SrVO_3$ results in a dramatic change of its physical properties. The $^{51}V$-NMR spectrum of 15R-$SrVO_{2.2}N_{0.6}$ powder can be fitted with a sharp and a broad component (Fig. 2a). The former is centered at $\mu_0H(T) = 1.4270$ T, which is identical to $Sr_3V^{5+}_2O_8$ and hence assigned to the tetrahedral V1. This site has an extremely slow relaxation time of $1/T_1 = 0.06$ s$^{-1}$ (Fig. 2b), indicating that $3d$ electrons are totally absent for this electronic configuration. Further, the very sharp peak implies a "strict" $V1O_3N$ configuration though anion order within the $SrO_2N$ layer is not seen from the diffraction study. By contrast, the broad component centered at 1.4243 T (cf. 1.4245 T in $SrVO_3$) has a much faster relaxation rate ($1/T_1 = 10.55$ s$^{-1}$), and is assigned to the octahedral (V2, V3) sites.

These results indicate that the triple-octahedral layer is electronically well separated by the double-tetrahedral layer. This is verified through our density functional theory (DFT) calculations that show a 2D Fermi surface with two cylinder-like sheets (Fig. 3a, Supplementary Fig. 3 and Supplementary Note 2), in stark contrast to the precursor $SrVO_3$ with its 3D Fermi surface[41]. Closer inspection shows that there are three pairs of flat Fermi

surfaces, implying weak overlap between different $t_{2g}$ orbitals. A similar feature can be seen in the (111) $SrTiO_3$ surface 2D electron gas (2DEG)[42]. NMR spectra of the octahedral site exhibit splitting at 5 K, accompanied by a divergence in $1/T_1T$ (Figs. 3b, c). The ordered moment is as low as $0.01\mu_B$ (where the hyperfine coupling of $-7.4$ T $\mu_B^{-1}$ for $LiV_2O_4$ was used[41]), indicating itinerant antiferromagnetism. The magnetic origin of the transition is also observed in field-dependent electrical resistance of the nitridized $SrVO_3$ film on $SrTiO_3$ (111) (Fig. 3d). These experimental findings, together with theoretically obtained 2D Fermi surfaces, strongly suggest that the observed transition is a spin density wave (SDW) transition owing to the large nesting effect[43].

## Strain-induced defect layer switching

As demonstrated above, polycrystalline 15R-$SrVO_{2.2}N_{0.6}$ possesses $(111)_p$ defect planes with the fivefold periodicity (namely, the *ooott* sequence where "*o*" and "*t*" refer to octahedral and tetrahedral layers). In order to investigate the epitaxial strain effects on oxynitrides, we fabricated an epitaxial $SrVO_3$ thin film using a (111)-oriented substrate for subsequent ammonolysis. Surprisingly, thermal treatment ($NH_3$ gas, 600 °C) of the $SrVO_3$ film on an LSAT (111) substrate resulted in the creation of new anion-defect layers. Scanning transmission electron microscopy (STEM) clearly indicated a $(112)_p$ planar defect, as shown in Fig. 4a and Supplementary Fig. 4a for high-angle annular dark field (HAADF) and annular bright-field (ABF) images. The Fourier transform data shows sevenfold satellite peaks along $[112]_p$. Based on STEM and XRD along with simulation, we constructed a monoclinic structure 7M-$SrV(O,N)_{2.71}$ ($Sr_{14}V_{14}(O, N)_{38}$) with tetrahedra and pyramids around anion defects (Supplementary Note 3, Fig. 4a, c, e, Supplementary Fig. 4 and Supplementary Tables 5 and 6), meaning that the composition is slightly different from the bulk composition of $SrVO_{2.2}N_{0.6}$. A similar $(112)_p$ planar defect has recently been reported in $BaFeO_{2.33}F_{0.33}$ ($Ba_3Fe_3O_7F$) powder, but with a threefold periodicity[44]. Nuclear reaction analysis (NRA) and elastic recoil detection analysis (ERDA), respectively, gave the nitrogen content of $x = 0.8$ (2) and 0.54(3) in $SrVO_{2.71-x}N_x$. Note that the oxynitride films have threefold domain structures related by 120° rotation (Supplementary Note 3 and Supplementary Fig. 4h–j).

Despite extensive research on perovskite oxides, conversion of oxygen-vacancy planes via thin film fabrication has not been previously observed. Past studies on thin films have been exclusively limited to increasing random oxygen vacancies[18–21] or controlling the crystallographic orientation of known defect perovskites such as $Ca_2Fe_2O_5$ brownmillerite[22–27]. Given a lattice mismatch of −0.7% between 15R-$SrVO_{2.2}N_{0.6}$ and LSAT (111), it is naturally expected that the observed $(111)_p$ to $(112)_p$ switching arises as a result of the compressive biaxial strain from the substrate. For a proof-of-concept, we grew epitaxial $SrVO_3$ films on $LaAlO_3$ (111) (LAO) and $SrTiO_3$ (111) (STO) substrates and nitridized under the same condition (Supplementary Note 4). For the LAO (111) substrate with a nominal strain of −2.3%, although the film was partially relaxed to about −1.0%, we observed $(112)_p$ defect planes with a sevenfold superlattice (Supplementary Fig. 5), as in the case of LSAT substrate. On the contrary, when an STO (111) substrate with 0.2% tensile strain was used, we observed $(111)_p$ defect planes (Fig. 4b). These results strongly support that the vacancy-plane switching originates from substrate strain (Fig. 4g).

To rationalize the formation of 7M-$SrV(O,N)_{2.71}$ on $LaAlO_3$ (SVON-112) versus 15R-$SrVO_{2.2}N_{0.6}$ in the bulk (SVON-111), we used DFT to calculate the relative energies of the two SVON orientations at various biaxial lattice strains. For these calculations, we used the SCAN metaGGA functional[45], which is more accurate

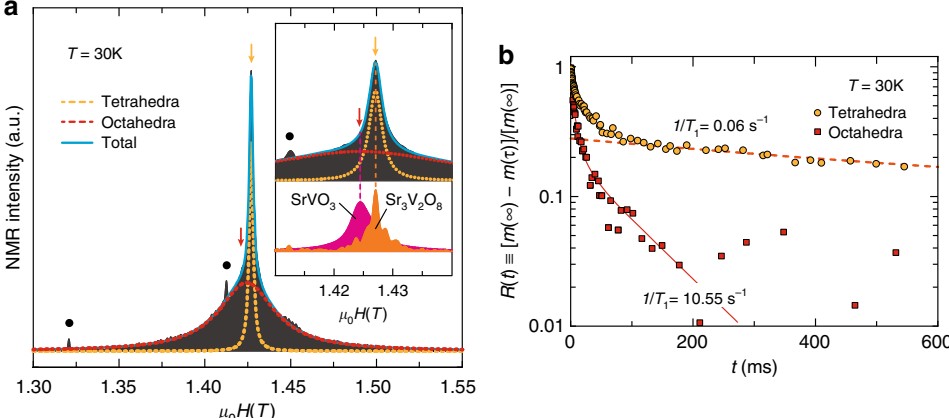

**Fig. 2 $^{51}$V-NMR for 15R-SrVO$_{2.2}$N$_{0.6}$. a** Field-swept NMR spectrum at 30 K showing peaks from tetrahedral (orange) and octahedral (red) sites. Solid circle indicates the signal from the Cu coil. The orange and red arrows indicate the positions at which $1/T_1$ was measured. Inset is an enlarged plot, where spectra of SrVO$_3$ and Sr$_3$V$_2$O$_8$ are shown for comparison. **b** Time after saturation pulse dependence of relaxation of $R(t) \equiv [m(\infty) - m(t)]/m(\infty)$, where $m$ represents nuclear magnetization at a time $t$ after a saturation pulse, at the tetrahedral (orange) and octahedral (red) sites. The lines are the fit to extract the time constant. Tetrahedral data with $t < 100$ ms were not considered due to octahedral-site contribution (see yellow arrow in **a**).

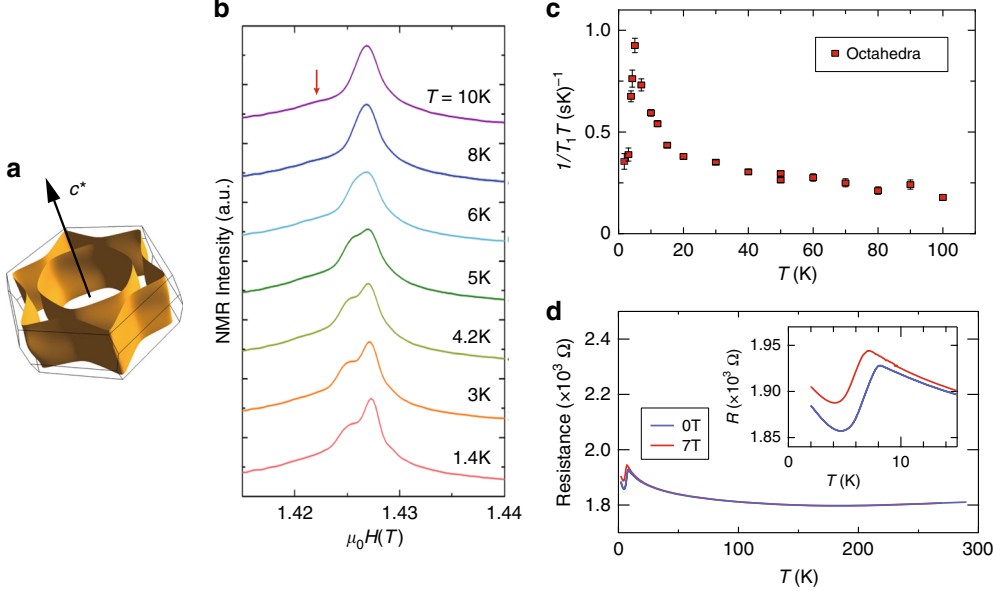

**Fig. 3 Physical properties of the (111)$_p$ defect structure. a** Calculated Fermi surface for 15R-SrVO$_{2.2}$N$_{0.6}$. **b** $^{51}$V-NMR spectra for SrVO$_{2.2}$N$_{0.6}$ at low temperatures. The ordered moment is estimated to be <0.01$\mu_B$. **c** Temperature dependence of $1/T_1T$ at the octahedral site (red arrow in **a**), indicating a magnetic transition at 5 K. Error bars are standard deviation. **d** Temperature dependence of electrical resistance of the nitridized SrVO$_3$ film on SrTiO$_3$ (111) at 0$T$ and 7$T$, characteristic of weak antiferromagnetism due to SDW transition at around 8 K.

in computing thermochemical properties than the Perdew–Burke–Ernzerhof (PBE) functional[46,47]. For SVON-112, we assumed a plausible distribution of N and O (to give Sr$_7$V$_7$O$_{14}$N$_5$) with reference to SVON-111 (Supplementary Note 5). The free energy of strained SVON is modeled using a thermo-dynamic grand potential[45], with chemical potentials of $\mu_N$ and $\mu_O$ set to approximate NH$_3$(g) flow rates at 200 mL min$^{-1}$ at 600 °C, as benchmarked by Katsura et al. (Supplementary Fig. 9)[46]. Figure 4f and Supplementary Fig. 10 demonstrate that the SVON-112 structure is stabilized when compressive isotropic biaxial strain is applied, with the minimum energy at the position close to the La–La distance of LaAlO$_3$ ($d_{La–La} = 5.390$ Å), consistent with the experiment. On the other hand, at the Sr–Sr distance of SrTiO$_3$ ($d_{Sr–Sr} = 5.579$ Å), the minimum free energy SVON compound is SVON-111. Further details of the calculation can be found in Supplementary Note 5.

Further insights are garnered from observation that the oxynitride film grown on SrTiO$_3$ (111) exhibits sixfold superlattice reflections in TEM and XRD (Fig. 4b and Supplementary Fig. 6), implying that the defect plane appears in every six SrO$_3$ layer, namely, the *oooott* sequence (Fig. 4d), in contrast to the $(ooott)_3$ sequence in the bulk (Fig. 1c). This observation suggests that even small strains (0.2%) may affect the structure, offering fine and extensive tuning of not only the direction but also the periodicity (density) of anion-defect layers (Fig. 4g), which will alter the chemical and physical properties. The slightly higher SDW transition temperature of 8 K for this film (Fig. 3d) compared with the bulk (5 K) might also be related to the periodicity change. We carried out similar reactions to the SrVO$_3$ films grown along [001] using LSAT and LaAlO$_3$ substrates. However, these films were nitridized without forming superlattices (Supplementary Figs. 7 and 8), indicating that the strain direction, or how the VO$_6$

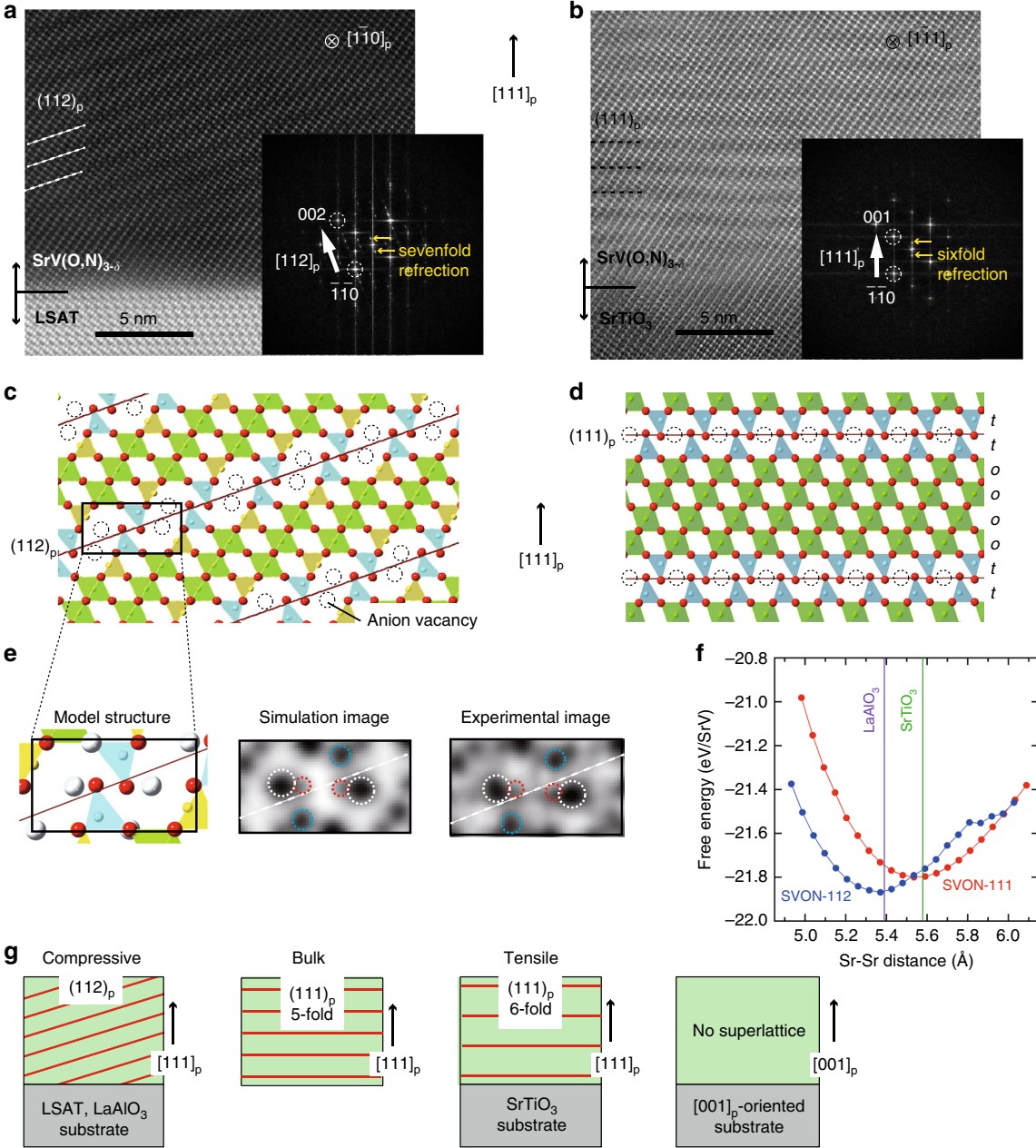

**Fig. 4 Strain-induced switching of anion-vacancy layers in [111]-oriented oxynitride thin films. a**, **b** Cross-sectional HAADF image of the nitridized SrVO₃ film on **a** LSAT (111) and **b** SrTiO₃ (111) interfaces taken along [1̄10] and their Fourier transform patterns. **c**, **d** The corresponding structures with vacancies along (**c**) (112)$_p$ and (**d**) (111)$_p$ with the *oooott* sequence (see Fig. 1c for the bulk). Sr atoms are omit**t**ed. **e** Experimental and simulated ABF images of the structure in **c**. **f** Schematic view of biaxial strain-induced switching of anion-vacancy layers. Red lines represent vacancy layers. **g** Thermodynamic competition between SVON-111 and -112 as a function of biaxial strain. DFT-calculated free energies of SVON-111 and SVON-112 under flowing NH₃(g) at 600 °C, as a function of the Sr−Sr distance. The corresponding cation-cation distances for LaAlO₃ and SrTiO₃ substrates are also given.

octahedron is deformed greatly affects the vacancy-layer formation (or not). Recent first-principles study on LaAlO₃ showed epitaxial strain in the (001) and (111) planes results in distinct phases with different octahedral rotation patterns[47].

The observed periodicity change may also be subject to kinetic aspects. It is shown that brownmillerite CaFeO₂.₅ films with tetrahedral/octahedral layers stacked parallel and perpendicular to a substrate display distinct reactivities with anisotropic oxygen diffusions[25]. Upon the reduction of CaFeO₂.₅ films to CaFeO₂, oxide anions are prone to migrate perpendicular to the substrate. Our oxynitride films on STO (111) are overall epitaxial but the superlattice is slightly corrugated (Fig. 4b) unlike films with (112)$_p$

defect planes (Fig. 4a). Such corrugation might arise from faster oxygen extraction and nitrogen insertion perpendicular to the substrate and also be a cause for the elongated periodicity from five- to sixfold. If this is the case, the reaction temperature and time could also be parameters to tune anion-vacancy order patterns. It would be of further interest to see if—or how—the film thickness also plays a role in the resulting lattice periodicity/structure.

In the last two decades, there has been tremendous progress in the study of artificial perovskite superlattice of at least two sets of perovskite compositions (SrTiO₃/LaAlO₃, LaMnO₃/SrMnO₃, etc.), leading to the discovery of novel phenomena such as superconductivity and quantum Hall effects in 2DEG[48–51]. Given

the abundance of perovskite oxides forming linear and planar anion defects, it is clear that the present study offers exciting new opportunities to design perovskite superlattices by chemically modifying simple 3D perovskite oxides with controllable anion-vacancy planes through the lattice strain in order to obtain unique properties and phases such as 2DEG. Furthermore, in an even broader perspective not just limited to oxides, the vacancy engineering we demonstrated here might also be utilized for phonon control and to strongly modify thermal transport properties given less controlled approaches to yield record thermoelectric performance in commercial materials such as $B_{0.5}Sb_{1.5}Te_3$ and $PbSe$[52,53].

## Methods

**Preparation of powder samples.** $Sr_2V_2O_7$ was prepared by solid state reactions. Stoichiometric mixtures of $SrCO_3$ (99.9%; High Purity Chemicals) and $V_2O_5$ (99.99%; Rare Metallic) were ground, pelletized, and preheated at 900 °C in air for 48 h with intermediate grindings. $SrVO_3$ was obtained by annealing the $Sr_2V_2O_7$ pellet at 900 °C in flowing $H_2$−$Ar$ (10:90 vol%) gas at a rate of 100 mL min$^{-1}$ for 24 h. $SrVO_2H$ was prepared by a topochemical reaction of $SrVO_3$ with $CaH_2$ (99.99%; Sigma-Aldrich) as described elsewhere[32]. Ammonolysis reactions were performed for $SrVO_2H$, $SrVO_3$, and $Sr_2V_2O_7$; the powder samples on a platinum boat were placed in a tubular furnace (inside diameter of 26 mm) and were reacted under $NH_3$ flow (200 mL min$^{-1}$) at various temperatures (300–800 °C) for 12 h, at a heating and cooling rate of 100 °C h$^{-1}$. Before reaction, the tube was purged with dry $N_2$ gas at ambient temperature to expel oxygen and moisture.

**Characterization of powder samples.** We conducted XRD measurements using a D8 ADVANCE diffractometer (Burker AXS) with a Cu $K_\alpha$ radiation. The nitrogen contents of the samples after different $NH_3$ treatments were determined by the combustion method (elemental analysis) at Analytical Services, School of Pharmacy, Kyoto University. Approximately 2 mg was used for each experiment, and three data sets were averaged. A high-resolution synchrotron XRD experiment for the nitridized samples was performed at room temperature using the large Debye–Scherrer cameras with a semiconductor detector installed at the beamline BL02B2 (JASRI, SPring-8). Incident beams from a bending magnet were monochromatized to $\lambda = 0.42073(1)$ and $0.78086(2)$ Å for the $SrVO_3$ samples after $NH_3$ treatment at 300 and 600 °C, respectively. Finely ground powder samples were sieved through a 32-μm mesh sieve and were packed into Pyrex capillaries with an inner diameter of 0.2 mm, which were then sealed. The sealed capillary was rotated during measurements to improve randomization of the individual crystallite orientations. Powder neutron diffraction data were collected at room temperature using the high-resolution powder diffractometer BT-1 ($\lambda = 1.54060$ Å) at the NIST Center for Neutron Research (NCNR), and time-of-flight (TOF) diffractometers, iMATERIA[54], and NOVA at the Japan Proton Accelerator Research Complex (J-PARC). The collected synchrotron and neutron profiles were analyzed by the Rietveld method using RIETAN-FP program[55] (SPring-8 and NIST) and Fullprof program[56] (J-PARC).

**Preparation of thin film samples.** Epitaxial $SrVO_3$ thin films with a thickness of 100 nm were deposited on single crystalline substrates of $SrTiO_3$ (111), $(LaAlO_3)_{0.3}(SrAl_{0.5}Ta_{0.5}O_3)_{0.7}$ (LSAT) (111), $LaAlO_3$ (111), (LSAT) (100), and $LaAlO_3$ (100), using the pulsed laser deposition technique. We used a KrF excimer laser at $\lambda = 248$ nm, with a deposition rate of 2 Hz and an energy of ~2 J cm$^{-2}$. The substrate temperature was kept at 950 °C during the deposition, with an oxygen partial pressure of $5 \times 10^{-7}$ Pa. Subsequently, the oxide films were placed in a platinum boat and heated under an $NH_3$ flow at 600 °C (LSAT (111)) and 620 °C ($SrTiO_3$ (111), $LaAlO_3$ (111), LSAT (100), and $LaAlO_3$ (100)) for 12 h, in the same manner as in the case of powder samples.

**Characterization of thin films.** Structures of the films were examined by XRD with Cu $K_\alpha$ radiation (Bruker AXS D8 DISCOVER), where the diffractometer in parallel beam geometry was equipped with one- and two-dimensional detectors. STEM images were acquired using an aberration-corrected microscope (Titan cubed, FEI) operating at an acceleration voltage of 300 kV. The convergence semiangle of the incident probe was 18 mrad, while the detector collection semi-angles ranged from 77 to 200 mrad for HAADF and 10–19 mrad for ABF imaging. The high-resolution STEM images in Fig. 4 and Supplementary Fig. 4 were Fourier filtered for noise reduction. A thin specimen for the STEM observation was prepared using a focused ion beam (FIB) instrument (FB-2000, Hitachi). A low-energy Ar-ion milling (NanoMill Model1040, E.A. Fischione Instruments) was performed after the FIB processing to eliminate surface damaged layers.

The amount of nitrogen in the oxynitride film was evaluated by NRA method using the $^{15}N(p,\alpha\gamma)^{12}C$ resonant nuclear reaction at 899.98 keV. NRA measurements were carried out with a 1-MV tandetron accelerator at Tandem Accelerator Complex, University of Tsukuba. The experimental error of around

±20% originates from instability of accelerator and small thickness of the film. Details of the measurement and calibration are given in ref. [17].

The anion compositions of the films were also determined by ERDA with a 38.4 MeV Cl beam generated by a 5-MV tandem accelerator (Micro Analysis Laboratory, The University of Tokyo [MALT])[57]. The experimental errors were around 5% under a typical condition.

**Physical property measurements.** A conventional spin-echo technique was used to measure $^{51}V$-NMR for the powder sample of 15R-$SrVO_{2.2}N_{0.6}$. $^{51}V$-NMR spectra (the nuclear spin of $I = 7/2$ and the nuclear gyromagnetic ratio of $^{51}\gamma/2\pi = 11.193$ MHz T$^{-1}$) were obtained by sweeping magnetic field in a fixed frequency of 15.9 MHz. Nuclear spin–lattice relaxation rate $1/T_1$ was determined by fitting the time variation of the nuclear magnetization $m(t)$ after a saturation pulse to a theoretical function for a nuclear spin $I = 7/2$ relaxation at the central transition. The electrical resistance of the nitridized $SrVO_3$ film on $SrTiO_3$ (111) was measured down to 2 K by means of a standard four-probe method using a physical property measuring system (Quantum Design, PPMS) at 0 and 7 T.

**DFT calculations.** Crystal structure optimization for 15R-$SrVO_{2.2}N_{0.6}$ was performed by using the PBE parametrization of the generalized gradient approximation[58] and the projector augmented wave method (PAW)[59] as implemented in the Vienna ab initio simulation package (VASP)[60–63]. In the PAW potentials used in this study, the following states are treated as core for each element: [Ar] $3d^{10}$ for Sr, [Ne] for V, and [He] for O and N, which is in common with the strain calculation described in the next paragraph and Supplementary Note 5. In order to obtain further information of the electronic structure, we performed band-structure calculation using the WIEN2k code[64] and then extracted the vanadium-$d$ Wannier functions from the calculated band structures using the Wien2Wannier and Wannier90 codes[65–68]. For this purpose, we also used the PBE functional. Further details of these calculations are discussed in Supplementary Note 2.

For the biaxial strain calculations, ordered approximants for 7M-$SrV(O,N)_{2.71}$ and 15R-$SrVO_{2.2}N_{0.6}$ were generated in *pymatgen*[69]. Stability calculations were performed using VASP, with the SCAN metaGGA functional[70]. The relative free energies of the 7M and 15R phase for a given temperature, ammonia flow rate, and biaxial strain, are calculated using a thermodynamic grand potential, $\Phi$, as

$$\Phi_{SVO_xN_y}(\epsilon_{2D}, \mu_O, \mu_N) = G_{SVO_xN_y} - \mu_O x_O - \mu_N x_N,$$

where the chemical potentials, $\mu$, of oxygen and nitrogen are determined from benchmarked nitrogen activities by Katsura[46]; further details of this referencing is discussed in Supplementary Note 5. For both SVON (111) and (112) orientations, we first relax the unit cell until forces are converged to $10^{-8}$ eV Å$^{-1}$, and then we perform the slab transformation into (111) or (112) Miller indices and apply isotropic biaxial strain.

## Data availability
The data that support the findings of this study are available from the corresponding author upon reasonable request.

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

## Acknowledgements

This work was supported by JSPS KAKENHI (JP16H06438, JP16H06439, JP16H06440, JP16H06441, JP16K21724, JP17H05481, JP18H01860, 20H00384), CREST (JPMJCR1421), and Project of Creation of Life Innovation Materials for Interdisciplinary

and International Researcher Development of MEXT, Japan. W.S. was supported by funding from the US Department of Energy, Office of Science, Basic Energy Sciences, under contract no. UGA-0-41029-16/ER392000 as a part of the Department of Energy Frontier Research Center for Next Generation of Materials Design: Incorporating Metastability. The synchrotron XRD experiments were performed at the BL02B2 of SPring-8 with the approval of JASRI. The ND experiment was performed at J-PARC and the NIST Center for Neutron Research. We thank the sample environment group at J-PARC MLF for their support on the neutron scattering experiments. We appreciate Mr. Satoshi Ishii, Prof. Kimikazu Sasa, and Dr. Hiroshi Naramoto of Univ. Tsukuba for their assistance in the NRA measurements. Certain commercial equipment, instruments, or materials are identified in this document. Such identification does not imply recommendation or endorsement by the National Institute of Standards and Technology nor does it imply that the products identified are necessarily the best available for the purpose.

## Author contributions

T.Y. and H.K. designed the research. T.Y., N.I., T.N., and F.T. synthesized and characterization of the bulk samples. T.Y., A.C., N.I., H.T., T. Maruyama, M.N., T.N., and T. Hasegawa synthesized and characterization of the thin film samples. S.Y., T. Mori, and K. Kimoto performed STEM experiments. S.K. and K. Ishida performed NMR measurements. M.O. and K. Kuroki performed DFT calculations on the bulk, while W.S. on the strained films. K.F., M.Y., C.M.B., T. Honda, K. Ikeda, and T.O. collected neutron data. M.S. and Y.H. conducted NRA measurements. Y.S., Y.H., and D.S. performed ERDA. All the authors discussed the results. T.Y. and H.K. wrote the manuscript, with comments from all the authors.

## Competing interests

The authors declare no competing interests.
