## [Peer Review File · Nature Communications]

REVIEWER COMMENTS

Reviewer #1 (Remarks to the Author):

The authors present a combined experimental and theoretical study, investigating the formation of planar defects in nitridated perovskite SrVO₃. They show that in the polycrystalline oxynitride an anion vacancy layer forms every 5 layers along the [111] direction, reminiscent of the SrCrO_{2.8} structure. They show that higher oxidation states of V are found adjacent to the vacancy layer and that the layered crystal structure leads to a 2D electronic structure. Moreover they investigate the nitridation of compressive and tensile strained SrVO₃ thin films on LaAlO₃ and SrTiO₃ substrates respectively and show that strain can switch the anion vacancy plane to (112) under compression, while tensile strain affects the periodicity of (111) anion vacancy planes. The authors speculate on the properties of these novel phases for electronic applications.

While as a theoretician I cannot give a fully qualified evaluation, I perceive the experiments as well conducted. The theoretical work presented in the manuscript, however leaves much to wish for in terms of both exhaustiveness of sampling as well as a lack of detail as I will further detail below. Due to these shortcomings I cannot recommend the manuscript for publication in Nature Communications, as for a journal of this caliber, I expect the authors to leave no rock unturned and present a complete and fully exhaustive theoretical analysis of the nitridated SrVO₃ phases.

1) DFT calculations were performed with three different setups. In VASP with PBE, in Wien2k without specifying the functional and in VASP with SCAN. Why this zoo of methods? How well do the different methods agree? At present the DFT results seem inconsistently thrown together and are therefore not of much value. Also how well does the Wannier function-based tight-binding model reproduce the full DFT band structure and which states were included in its parametrization?

2) What valence states were treated in the VASP PAW potentials? Why should a 6x6x6 k-mesh be good choice for the elongated 15R-SrVO_{2.2}N_{0.6} structure?

4) How was biaxial strain applied (unequal lengths of cell vectors and potentially angles different from 90°) and how were the strained structures relaxed. It is mentioned that the unstrained lattice parameters were relaxed at fixed cell shape. What residual stress does this leave in the cell?

5) While it is fairly established that perovskite oxynitrides prefer a cis order, I would have expected the authors to carry out a more exhaustive sampling of the anion order, in particular in view of the presence of the vacancy layer that breaks the symmetry and also as a function of strain since the anion order was shown to be depend on that parameter.

6) In supplementary note 5, an approach is outlined to estimate the chemical potentials required to compare the total energies of the (111) and (112) layered structures that have different stoichiometry. I find the extrapolation of the activities to higher flow rates (supplementary Fig. 9) extremely handwaving. As the interdependent chemical potentials will strongly affect the agreement with experiment shown in figure 4g, the authors need to quantify the evolution of the Sr-Sr distance where both phases coexist as a function of both the nitrogen and oxygen chemical potential in sensible ranges. Also how transferable is the employed data for nitridation of metals to the nitridation of an oxide?

Minor points:

1) The first two sentences of the abstract should be revised as they are really hard to understand (not ideal for an abstract)

2) What does "commensurate with respect to the underlying lattice" on page 3 mean? The lattice is a collection of points corresponding to integer multiples of the cell vectors without any knowledge of the atomic structure and hence a vacancy concentration cannot be commensurate with the lattice.

3) I am not sure "nitridized" is a proper adjective. I am more used nitridated but leave this up to the authors.

4) The Fermi surface shown in figure 1d seems a bit out of context and it is also wrongly referenced as "supplementary figure 3" on page 7.

5) On page 6 the authors speculate on anion diffusion, but I can't understand what they try to say with the following sentence: "In this regard, the diffusion of the nitride ions might be promoted by "in-situ" formation of SrO₂ layers during the ammonolysis reaction at 600 °C." Maybe this could be reworded.

6) In supplementary note 2, there is the following sentence: "The unit cell vectors were changed in the band-structure calculation by the WIEN2k code as shown in Supplementary Fig. 3a." I cannot understand what this is supposed to mean.

7) There are small grammar errors throughout the manuscript that should be corrected.

Reviewer #2 (Remarks to the Author):

This work is well done and belongs to the highest quality reports, in the criteria of novelty of the subject, integrity of the methodology, and scientific discussion. However, I would like the authors consider the following points in the future revision.

1. Upon the ammonolysis at 300 degC, SrVO₂H turned into SrV_{0.7}N_{0.2}, where the increase of oxygen content is non-negligible. What is the possible source of oxygen?

2. SrVO₂H and SrVO₃ are converted to similar phases by the ammonolysis at 600 degC. In this regard, the experimental result of SrVO₂H is limitedly informative. In addition, I wonder if the authors checked the stability of 15R phase, by either an ammonolysis at higher temperature than 600 degC or by TGA.

3. Nowhere was mentioned the color of powder samples (my apology if I missed).

Reviewer #3 (Remarks to the Author):

Designing novel crystalline structure with emergent properties forms a central point for solid state chemistry and condensed matter physics study. In this paper, Yamamoto et al., reported a new strategy to achieve perovskite oxynitrides with ordered anion vacancy, and to support their claim, they have carried out extensive structural and compositional characterizations and the results are convincing. Over all, the results would be of great interest to a wide range community, especially motivated by the study of layered oxide for novel electronic state as well as perovskite oxynitride for catalysis. Although the crystalline structural analysis is comprehensive and solid, the referee found that the properties characterization is premature.

1) NMR in Fig. 2. The authors fitted the data with two experimental decay functions to obtain corresponding time constants. However, the referee noticed that the authors omitted the data with $t < 100$ ms especially for the tetrahedra. Can the authors explain the reason?

2) The wide range of temperature less-dependent RT is very puzzling. Can the authors elaborate further? Also the smoking gun evidence for the SDW is missing.

- 3) The argument of domain-free single-crystalline in page 8 seems not well-defined, since the films should be multi-domain considering the stacking period and vacancy orientation, and this should be stated precisely.
- 4) It would be interesting/ important to know the thermal stability of the formed phases.

Reviewer #1

The authors present a combined experimental and theoretical study, investigating the formation of planar defects in nitridated perovskite SrVO₃. They show that in the polycrystalline oxynitride an anion vacancy layer forms every 5 layers along the [111] direction, reminiscent of the SrCrO_{2.8} structure. They show that higher oxidation states of V are found adjacent to the vacancy layer and that the layered crystal structure leads to a 2D electronic structure. Moreover they investigate the nitridation of compressive and tensile strained SrVO₃ thin films on LaAlO₃ and SrTiO₃ substrates respectively and show that strain can switch the anion vacancy plane to (112) under compression, while tensile strain affects the periodicity of (111) anion vacancy planes. The authors speculate on the properties of these novel phases for electronic applications.

We would like take this opportunity to thank the referee for his/her summary of our work.

While as a theoretician I cannot give a fully qualified evaluation, I perceive the experiments as well conducted. The theoretical work presented in the manuscript, however leaves much to wish for in terms of both exhaustiveness of sampling as well as a lack of detail as I will further detail below. Due to these shortcomings I cannot recommend the manuscript for publication in Nature Communications, as for a journal of this caliber, I expect the authors to leave no rock unturned and present a complete and fully exhaustive theoretical analysis of the nitridated SrVO₃ phases.

We thank the referee again for his/her constructive criticism and nice suggestions regarding the theoretical part. In the following, we give a response to each of your comments, combined with modification of the manuscript and Supporting Information. We sincerely hope that these improvements detailed below will satisfy your requirements and those pointed out by other reviewers.

1) DFT calculations were performed with three different setups. In VASP with PBE, in Wien2k without specifying the functional and in VASP with SCAN. Why this zoo of methods? How well do the different methods agree? At present the DFT results seem inconsistently thrown together and are therefore not of much value. Also how well does the Wannier function-based tight-binding model reproduce the full DFT band structure and which states were included in its parametrization?

Wien2k is an all-electron DFT code, and is better poised for electronic band structure calculations, while the structural optimization is basically not capable in this code and so we employed VASP for such purpose. As for the choice of energy functionals, SCAN is currently the best DFT functional for thermochemical properties [1,2], and so we employ SCAN to calculate the thermodynamic analyses in this paper. For other analyses, we consistently used the PBE functional, which is well tested for general purposes. We have added a description that PBE was used in WIEN2k calculation in Method section. Although this looks like a 'zoo' of methods, in fact we are choosing the best code and functional for each specific task at hand. A difference in calculation results caused by the use of different softwares has been very carefully examined, e.g., in Ref. [3]. It is known that VASP (PAW) and WIEN2k generally show very good consistency. Thus, we believe that the use of the two softwares (VASP and WIEN2k) does not affect the conclusion of our study.

The Wannier interpolation scheme was used for plotting the Fermi surface with a very fine k -mesh ($100 \times 100 \times 100$). For this purpose, as pointed out by Reviewer #1, it is crucial whether the original DFT band structure is well reproduced by the Wannier-function-based tight-binding model. We have added a new figure (Supplementary Figure 3c) showing the band structures calculated with the Wannier-function-based tight-binding model and DFT. These two band structures agree well near the Fermi level, which validates the use of the Wannier functions. For these calculations, Wannier functions of Vanadium- d orbitals were extracted as described in Method section. To improve clarity, we have changed the following sentence in Supplementary Note 2, "... first-principles band-structure calculation and the subsequent Wannier construction were performed..." to "... first-principles band-structure calculation and the subsequent Wannier construction for the V- d orbitals were performed...".

[1] J. Sun, *et al.*, *Nat. Chem.* 8, 831 (2016).

[2] J. Yang *et al.*, *Phys. Rev. B* 100, 035132 (2019).

[3] K. Lejaeghere *et al.*, *Science* 351, aad3000 (2016).

2) What valence states were treated in the VASP PAW potentials? Why should a 6x6x6 k-mesh be good choice for the elongated 15R-SrVO_{2.2}N_{0.6} structure?

We have added the following description in Method section "In the PAW potentials used in this study, the following states are treated as core for each element: [Ar] 3 d^{10} for Sr,

[Ne] for V, and [He] for O and N, which is in common with the strain calculation described in the next paragraph and Supplementary Note 5.”

As written in Supplementary Note 2, we used the unit cell vectors $a_1 = (a/2, -(\sqrt{3}/2)a, c/3)$, $a_2 = (a/2, (\sqrt{3}/2)a, c/3)$, $a_3 = (-a, 0, c/3)$, for VASP calculation of $15R\text{-SrVO}_{2.2}\text{N}_{0.6}$, where the length of the corresponding three reciprocal vectors are the same. Therefore, we took the same number of k -points for each direction (i.e., $6 \times 6 \times 6$ k -mesh). It can be confusing to the reader that the definition of the unit cell vectors is different between VASP and WIEN2k because WIEN2k changes the unit cell vectors automatically. Thus, we have placed the description of the k -mesh and the unit cell vectors for each software at the same place (Supplementary Note 2), which were presented separately in the original manuscript (Method and Supplementary Note 2). We believe that this change improves the clarity of the manuscript.

4) How was biaxial strain applied (unequal lengths of cell vectors and potentially angles different from 90°) and how were the strained structures relaxed. It is mentioned that the unstrained lattice parameters were relaxed at fixed cell shape. What residual stress does this leave in the cell?

For both SVON (111) and (112) orientations, we first relax the unit cell until forces are converged to 10^{-8} eV/Å, then we perform the slab transformation into (111) or (112) Miller indices and apply isotropic biaxial strain. In other words, we scale the 2D lattice by area, and then we report the Sr–Sr distance with its relationship to this 2D isotropic scaling. We have included this information into the description of the DFT methods.

We thank Reviewer #1 for bringing up this important point of unit cell stresses. Below, we plot the stresses in the SVON (111) and (112) unit cell on the same axes as Figure 4g. We find that at the LaAlO_3 and SrTiO_3 lattice parameters, the unit cell stresses for SVON-112 and SVON-111 are near zero, respectively. This is what we anticipate from Figure 4g, as we know that $\sigma = dE/d\epsilon$ (to a first-order approximation). At the bottom of the E - ϵ parabolas, $dE/d\epsilon = 0$, meaning that for the Sr–Sr distance of LaAlO_3 , the unit cell stress for SVON-112 should be about zero, and the unit cell stresses for SVON-111 should also be near zero, which is indeed what we measured from the calculations. We have included the figure below in Supplementary Figure 10.

Unit cell stresses in the SVON-111 and SVON-112 structures as a function of the Sr–Sr distance, as referenced against Figure 4g. The SVON unit cells are strained isotropically in the 2D biaxial directions.

Reviewer #1 asks about the biaxial strain with unequal lengths of cell vectors and potentially angles different than 90° . We anticipate that Reviewer #1 is asking about the local atomistic stresses that may form at the heteroepitaxial interface between the perovskite substrate and SVON. However, the local stresses and strains are strongly dependent on the atomistic structure and bonding coherency of this heteroepitaxial interface. This was not resolved experimentally (and would be highly non-trivial to do so), and so we also do not compute it. However, we emphasize that bulk stresses in SVON would dominate the mechanical energy terms ($\sigma\epsilon$) of the heteroepitaxial structure, which should be referenced with respect to the bulk, unrelaxed SVON structure. Indeed, our analysis reproduces the qualitatively observed switching behavior of the SVON anion-vacancy ordering as a function of the LaAlO₃ vs. SrTiO₃ substrate. For this reason, we believe that our theoretical treatment of isotropic 2D lattice strains is appropriate.

5) While it is fairly established that perovskite oxynitrides prefer a cis order, I would have expected the authors to carry out a more exhaustive sampling of the anion order, in particular in view of the presence of the vacancy layer that breaks the symmetry and also as a function of strain since the anion order was shown to be depend on that parameter.

The presence of a huge number of anion configurations is one of the characteristics of mixed-anion compounds, but it is a challenging problem to calculate. For example, the number of possible anion configurations for the bulk 15R-SrVO_{2.2}NO_{0.6} exceeds one million if we consider the supercell to be only three times larger than our calculation, as described in the Supplementary Note 2. This computational work deals specifically with the role of substrate strain in switching the anion vacancy ordering in the observed SVON structures. For this reason, we used the experimentally-resolved SVON structures as the two input structures for our analysis. We agree with Reviewer #1 that a more thorough enumeration of ordered SVON structures that this would be an interesting direction, but that would go beyond the scope of this work (the first experimental report on this phenomenon) and would be better suited to be covered in a future publication on anion vacancy order in oxynitrides.

6) In supplementary note 5, an approach is outlined to estimate the chemical potentials required to compare the total energies of the (111) and (112) layered structures that have different stoichiometry. I find the extrapolation of the activities to higher flow rates (supplementary Fig. 9) extremely handwaving.

We understand why Reviewer #1 finds the extrapolation of activities to higher flow rates unsatisfying, but we emphasize that this is the *best thermodynamic analysis that can be done with available data*. This is because, as discussed in Katsura [4], the ammonia fugacity at elevated flow rates behaves non-ideally. Without knowing the mechanism of non-ideality, it is not possible to calculate the NH₃ fugacity at high flow rates from first-principles. We must therefore rely on an experimental benchmark to approximate the fugacity at a given flow rate.

This approach of benchmarking the non-ideal chemical potential of a compound by comparing an experimentally-observed phase transition against a calculated thermodynamic boundary is actually a very standard approach. For example, Caskey *et*

al. [5] take a similar approach to benchmark the plasma-cracked nitrogen chemical potential in a reactive sputtering experiment.

We note that such 'corrections' to the DFT-calculated chemical potentials of gas-phase materials is very commonly used in the *ab initio* thermodynamics community, for example, see the following references [6, 7].

[4] M. Katsura, *J. Alloys Compd.* 182, 91 (1992). (Ref. 46 in the manuscript)

[5] C. M. Caskey *et al.*, *Mater. Horizons* 1, 424 (2014).

[6] L. Wang *et al.*, *Phys. Rev. B* 73, 195107 (2006).

[7] S. Grindy *et al.*, *Phys. Rev. B* 87, 075150 (2013).

As the interdependent chemical potentials will strongly affect the agreement with experiment shown in figure 4g, the authors need to quantify the evolution of the Sr-Sr distance where both phases coexist as a function of both the nitrogen and oxygen chemical potential in sensible ranges. Also how transferable is the employed data for nitridation of metals to the nitridation of an oxide?

The evolution of the phase co-existence is simply given by Lever Rule in this region, as is the case in all systems where a thermodynamic potential is plotted against an extensive thermodynamic variable (in this case, strain).

Also how transferable is the employed data for nitridation of metals to the nitridation of an oxide?

In principle, the data should be directly transferrable, because the phase whose chemical potential is being benchmarked is the ammonia gas phase. A properly-referenced gas phase chemical potential can be utilized for any thermodynamic analysis using this phase (whether it was benchmarked on nitridation of metals or oxides). This is, of course, assuming that Katsura [4] did a careful experimental measurement, minimizing any kinetic effects that might affect his chemical potential benchmarking. We can only trust and rely that Katsura [4] did their due diligence in this measurement. Again, we emphasize that this is the best available thermochemical data for this particular synthesis condition (flowing ammonia gas as a function of temperature and flow-rate).

Minor points:

1) *The first two sentences of the abstract should be revised as they are really hard to understand (not ideal for an abstract)*

We appreciate the reviewer for the advice.

“Perovskite oxides can host a variety of anion vacancies in the form of chains or layers, which results in various properties and thus the manipulation of vacancy patterns is of considerable interest. Separately, lattice strain between thin-film oxides and a substrate has been extensively exploited to improve functions or to induce novel states of matter.”

has been changed to

“Perovskite oxides can host a variety of anionic vacancy orders, which greatly change their properties, but the vacancy order pattern is still difficult to manipulate. Separately, lattice strain between thin film oxides and a substrate induces improved functions and novel states of matter, while little attention has been paid to changes in chemical composition.”.

2) *What does "commensurate with respect to the underlying lattice" on page 3 mean? The lattice is a collection of points corresponding to integer multiples of the cell vectors without any knowledge of the atomic structure and hence a vacancy concentration cannot be commensurate with the lattice.*

This sentence was indeed misleading. We have changed it to “If the anion vacancy concentration is a rational fraction of the oxygen stoichiometry in the unit cell,”.

3) *I am not sure "nitridized" is a proper adjective. I am more used nitridated but leave this up to the authors.*

While “nitridized” has 2,420 hits on google scholar. “nitridated” has 3330 hits.

Some examples of the former are in Refs. [8-13]. Refs. [12] and [13] are our previous work. Although “nitridated” is used slightly more than “nitridized”, we would like to use “nitridized” for consistency.

[8] A. Dabirian and R. v. d. Krol, *Chem. Mater.* 27, 708 (2015).

[9] H. Hajibabaei *et al.*, *Chem. Sci.* 7, 6760 (2016).

[10] Z. Wang, Can. Li *et al.*, *Adv. Energy Mater.* 6, 1600864 (2016).

[11] W. Sun, G. Ceder *et al.*, *Chem. Mater.* 29, 6936 (2010).

[12] T. Yajima *et al.*, *Nat. Chem.* 7, 1017 (2015). (Ref. 30 in the manuscript)

[13] R. Mikita *et al.*, *J. Am. Chem. Soc.* 138, 3211 (2016). (Ref. 31 in the manuscript)

4) The Fermi surface shown in figure 1d seems a bit out of context and it is also wrongly referenced as "supplementary figure 3" on page 7.

We agree with the reviewer's suggestion. This figure has been moved to Figure 3, which highlights the physical properties. We apologize for the incorrect reference to the figure. This has been fixed.

5) On page 6 the authors speculate on anion diffusion, but I can't understand what they try to say with the following sentence: "In this regard, the diffusion of the nitride ions might be promoted by "in-situ" formation of SrO₂ layers during the ammonolysis reaction at 600 °C." Maybe this could be reworded.

Recent studies on perovskite-based compounds [14-16] have shown that the AO₂ layer (1/3 oxygen depleted from the AO₃ layer) is the lattice with excellent oxide conductivity. From these facts, we deduced that the "in-situ" formed SrO₂ layer during the ammonolysis reaction can be a pathway for nitride-anion migration. The problem with the previous manuscript was that it did not specify the AO₂ layer as the promising anion conduction pathway. We modified the related sentences as follows:

"Although diffusion of nitride anions requires larger activation energy relative to oxide anion,³⁷ perovskite-based compounds with AO₂ layers have shown rapid oxide ion conduction.^{38,40} In this regard, the diffusion of the nitride ions might be promoted by "in-situ" formation of SrO₂ layers during the ammonolysis reaction at 600 °C."

[14] K. H. L. Zhang *et al.*, *Nat. Commun.* 5, 4669 (2014). (Ref. 38 in the manuscript)

[15] P. Ong, *et al.*, *J. Phys. Chem. Lett.* 8, 1757-1763 (2017). (Ref. 39 in the manuscript)

[16] S. Fop *et al.*, *J. Am. Chem. Soc.* 138, 16764-16769 (2016). (Ref. 40 in the manuscript)

6) In supplementary note 2, there is the following sentence: "The unit cell vectors were changed in the band-structure calculation by the WIEN2k code as shown in Supplementary Fig. 3a." I cannot understand what this is supposed to mean.

As mentioned in the Reply (2), the definition of unit cell vector is different between VASP and WIEN2k because WIEN2k changes the unit cell vectors automatically. While the WIEN2k adopts the unit cell vectors shown in Supplementary Fig. 3a, in VASP, we used the unit cell vectors $a_1 = (a/2, -(\sqrt{3}/2)a, c/3)$, $a_2 = (a/2, (\sqrt{3}/2)a, c/3)$, $a_3 = (-a, 0, c/3)$,

where a and c are shown in Fig. 1c in the main text. This can be confusing to the reader, so the revised manuscript contains the descriptions of the k -mesh and unit cell vectors for each software in Supplementary Note 2, which were presented separately in the original manuscript (Method and Supplementary Note 2). Also note that changing the unit cell vectors does not affect the partial DOS calculated by WIEN2k.

7) There are small grammar errors throughout the manuscript that should be corrected.

We carefully checked our manuscript and corrected errors.

Reviewer #2

This work is well done and belongs to the highest quality reports, in the criteria of novelty of the subject, integrity of the methodology, and scientific discussion. However, I would like the authors consider the following points in the future revision.

We are pleased to see that the reviewer highly evaluated our manuscript.

1. Upon the ammonolysis at 300 degC, SrVO₂H turned into SrV_{0.7}N_{0.2}, where the increase of oxygen content is non-negligible. What is the possible source of oxygen?

As pointed out in the Supplementary Note 1, the increase in oxygen content from the precursor SrVO₂H could be due to moisture contamination of the synthetic atmosphere.

2. SrVO₂H and SrVO₃ are converted to similar phases by the ammonolysis at 600 degC. In this regard, the experimental result of SrVO₂H is limitedly informative.

The synthesis of 15R-SrVO_{2.2}N_{0.6} followed by the discovery of strained induced phenomena is one of the highlights in this study. When this phase can be prepared by SrVO₃, the preparation of 15R-SrVO_{2.2}N_{0.6} from SrVO₂H looks limitedly informative, as the reviewer pointed out.

Nevertheless, we believe that it is very important to provide topochemical reactions as a background since low-temperature anion diffusion in solids is an important subject. In particular, there are two key findings for oxynitrides: (A) anion-exchange (H/N) reaction using BaTi(O,H)₃ oxyhydride at low temperatures (< 500 °C) [1] and (B) anion-exchange (O/N) reaction at moderate temperatures (< 800 °C) using oxyhydride EuTi(O,H)₃ [2]. Following [1, 2], we naturally we started our experiment by employing SrVO₂H as a

precursor for nitridization and obtained a cubic oxynitride at 500 °C (involving H/N exchange, i.e., (A)), and then 15R-SrVO_{2.2}N_{0.6} at 600 °C (involving O/N exchange, i.e. (B)). Later, we tested SrVO₃ as a 'control' experiment and unexpectedly obtained a less-nitridized cubic phase at 500 °C and 15R-SrVO_{2.2}N_{0.6} at 600 °C.

The previous manuscript dealt with (1) but not (2). Thus, the revised manuscript has been modified to readers to follow the history of research that led to the present findings.

[1] T. Yajima *et al.*, *Nat. Chem.* 7, 1017 (2015). (Ref. 30 in the manuscript)

[2] R. Mikita *et al.*, *J. Am. Chem. Soc.* 138, 3211 (2016). (Ref. 31 in the manuscript)

In addition, I wonder if the authors checked the stability of 15R phase, by either an ammonolysis at higher temperature than 600 degC or by TGA.

The 15R phase decomposes into Sr₂VO₃N and VN (non-perovskite phases) when heated above 800 °C in NH₃ atmosphere. Added a sentence "The 15R phase decomposes under NH₃ above 800 °C." in page 4. Reviewer #3 asked the similar question.

3. Nowhere was mentioned the color of powder samples (my apology if I missed).

You are not missing. Thank you for the suggestion. The sample is black. We added it to the main text (page 4).

Reviewer #3

Designing novel crystalline structure with emergent properties forms a central point for solid state chemistry and condensed matter physics study. In this paper, Yamamoto et al., reported a new strategy to achieve perovskite oxynitrides with ordered anion vacancy, and to support their claim, they have carried out extensive structural and compositional characterizations and the results are convincing. Over all, the results would be of great interest to a wide range community, especially motivated by the study of layered oxide for novel electronic state as well as perovskite oxynitride for catalysis. Although the crystalline structural analysis is comprehensive and solid, the referee found that the properties characterization is premature.

We would like to thank the reviewer to evaluate our study. We are pleased that the results would be of great interest to a wide range of communities.

1) NMR in Fig. 2. The authors fitted the data with two experimental decay functions to obtain corresponding time constants. However, the referee noticed that the authors omitted the data with $t < 100$ ms especially for the tetrahedra. Can the authors explain the reason?

The data for the tetrahedra (yellow circles in Fig. 2b) for $t < 100$ ms were omitted from the fitting because this region contains the contribution from the octahedra with fast relaxation. Please see Fig. 2a. It can be seen that the spectrum at 1.4270 T (indicated by the yellow arrow) includes contributions from both octahedral (V2, V3) and tetrahedral (V1) sites. Fitting the data (yellow circles) in the range of $0 < t < 15$ ms gave a relaxation rate $1/T_1 = 4.54 \text{ s}^{-1}$, which is comparable to 10.55 s^{-1} obtained from the fitting of the red squares. We have added a short description in the figure legend.

2) The wide range of temperature less-dependent RT is very puzzling. Can the authors elaborate further?

The temperature independent resistivity may be due to certain disorder effect, such as variable range hopping [1], weak Anderson localization [2,3], and impurity-induced electron-electron interaction [4]. While such effects increase resistivity upon cooling, the resistivity from intrinsic metal decreases, resulting in almost temperature independent electrical resistivity. Similar behavior has been observed with films and wires of metallic compounds [5]. Since our SVON thin film was processed under NH_3 gas flow, the surface of the film can be slightly damaged. This was added in the revised manuscript.

[1] N. F. Mott, *Philos. Mag.* 19, 835 (1969).

[2] E. Abrahams *et al.*, *Phys. Rev. Lett.* 42, 673 (1979)

[3] P. W. Anderson *et al.*, *Phys. Rev. Lett.* 43, 718 (1979).

[4] B. L. Altshuler *et al.*, *Phys. Rev. Lett.* 44, 1288 (1980).

[5] P. A. Lee and T. V. Ramakrishnan, *Rev. Mod. Phys.* 57, 287 (1985).

Also the smoking gun evidence for the SDW is missing.

As described in the original manuscript, itinerant antiferromagnetism and its phase transition were observed using NMR and field-dependent electrical resistance. Since SDW is one of the characteristics of antiferromagnetic order in itinerant electron systems as found in chromium [6-8], these experimental findings, together with theoretically

obtained 2D Fermi surfaces, strongly suggest that the observed phase transition originates from SDW.

We deduce that determining the spin structure by neutron diffraction is smoking gun evidence. However, the very small ordered moment ($0.01 \mu_B$ estimated from NMR) does not allow the observation of magnetic peaks even using a high-resolution time-of-flight neutron diffraction. In fact, our 2 K neutron diffraction collected at 20 K and 2K cannot detect magnetic reflections (see the figure below).

In the original manuscript, the SDW transition was initially inferred from the DFT data, followed by experimental results. In the revised manuscript, the experimental data are first presented to show itinerant antiferromagnetic order, and then, together with the DFT data, we concluded that SDW is likely.

[6] W. M. Lomer, *Proc. Phys. Soc.* 80, 489 (1962).

[7] S. Asano and J. Yamashita, *J. Phys. Soc. Jpn.* 23, 714 (1967).

[8] E. Fawcett, *Rev. Mod. Phys.* 60, 209 (1988). (Ref. 43 in the manuscript)

Time-of-flight powder neutron diffraction patterns for $\text{SrVO}_{2.2}\text{N}_{0.6}$ at 20 K (red) and 2 K (black).

3) The argument of domain-free single-crystalline in page 8 seems not well-defined, since the films should be multi-domain considering the stacking period and vacancy orientation, and this should be stated precisely.

We agree with the reviewer who stated that the argument of domain-free single-crystalline in page 8 is inappropriate. As mentioned in Supplementary Note 3, the fabricated oxynitride films have the three-fold domain structures related by 120° rotation. In the revised manuscript, “In order to obtain domain-free single-crystalline oxynitrides,

...” was changed to “In order to investigate the epitaxial strain effects on oxynitrides, ...”. We added the sentence “Note that the oxynitride films have three-fold domain structures related by 120° rotation (Supplementary Figs. 4h, 4i, 4j and Supplementary Note 3)”.

4) It would be interesting/ important to know the thermal stability of the formed phases.

We appreciate the reviewer for the good suggestion. The 15R phase decomposes into Sr₂VO₃N and VN (non-perovskite phases) when heated above 800 °C in NH₃ atmosphere. Added a sentence “The 15R phase decomposes under NH₃ above 800 °C.” in page 4. Reviewer #2 asked the same question.

REVIEWER COMMENTS

Reviewer #1 (Remarks to the Author):

While the authors have addressed most of my previous comments in a satisfactory fashion, there are two points, where additional minor changes would be beneficial.

1) While it is very clear from the response why different codes and functionals were used, it would be beneficial for the reader to add the (in my eyes justifiable) rationale behind choosing SCAN for the thermochemical properties as compared to PBE for consistency with the rest of the computations.

2) The authors may have misunderstood what I meant with part of my previous comment 6. It is obvious that the lever rule applies for phase coexistence. What I meant was that while the location of the minima on the Sr-Sr distance axis is fixed, their position on the energy axis depends on the O and N chemical potentials because the phases differ in stoichiometry. Given the - somewhat unsatisfactory but as the authors justify best possible - uncertainty in determining both the N as well as the O (via the H) chemical potential, my question was to what extent this uncertainty can affect the 112 vs 111 phase balance and hence the agreement with experiment. This could be addressed for example by reproducing figure 4g with different H and N activities within an sensible uncertainty range.

Reviewer #2 (Remarks to the Author):

The issues raised in the 1st-round review were adequately addressed, and I feel that the revised version (R1) is acceptable for publication.

Reviewer #3 (Remarks to the Author):

The authors nicely replied my comments and now the paper should be ready for publication.

Reviewer #1 (Remarks to the Author):

While the authors have addressed most of my previous comments in a satisfactory fashion, there are two points, where additional minor changes would be beneficial.

1) While it is very clear from the response why different codes and functionals were used, it would be beneficial for the reader to add the (in my eyes justifiable) rationale behind choosing SCAN for the thermochemical properties as compared to PBE for consistency with the rest of the computations.

We thank Reviewer #1 with the suggestion and have included additional text to justify the use of SCAN for calculating the thermochemical properties. This is included on Page 10 of the revised manuscript, excerpted below:

“To rationalize the formation of 7M-SrV(O,N)_{2.71} on LaAlO₃ (SVON-112) versus 15R-SrVO_{2.2}N_{0.6} in the bulk (SVON-111), we used DFT to calculate the relative energies of the two SVON orientations at various biaxial lattice strains. For these calculations, we used the SCAN metaGGA functional,⁴⁵ which is more accurate in computing thermochemical properties than the PBE functional.^{46, 47}”

[45] Jianwei, S., Ruzsinszky, A. & Perdew, J. P. Strongly constrained and appropriately normed semilocal density functional. *Phys. Rev. Lett.* **115**, 036402 (2015).

[46] Jianwei, S. *et al.* Accurate first-principles structures and energies of diversely bonded systems from an efficient density functional. *Nat. Chem.* **8**, 831-836 (2016).

[47] Yang, J. H., Kitchaev, D. A. & Gerbrand Ceder, G. Rationalizing accurate structure prediction in the meta-GGA SCAN functional. *Phys. Rev. B* **100**, 035132 (2019).

2) The authors may have misunderstood what I meant with part of my previous comment 6. It is obvious that the lever rule applies for phase coexistence. What I meant was that while the location of the minima on the Sr-Sr distance axis is fixed, their position on the energy axis depends on the O and N chemical potentials because the phases differ in stoichiometry. Given the - somewhat unsatisfactory but as the authors justify best possible - uncertainty in determining both the N as well as the O (via the H) chemical potential, my question was to what extent this uncertainty can affect the 112 vs 111 phase balance and hence the agreement with experiment. This could be addressed for example by reproducing figure 4g with different H and N activities within a sensible uncertainty range.

We apologize to Reviewer #1 for misinterpreting their previous comment. Indeed, we agree that the differences in oxygen and nitrogen stoichiometries between SVON-111 and SVON-112 could result in the errors from the nitrogen and hydrogen chemical potential plausibly influencing the equilibrium relationships between these two anion-vacancy orderings as a function of strain. We reanalyzed the free-energy figure (Figure 4g in the manuscript), and included error bars on the free-energy diagram corresponding to a range of nitrogen activities between $\log(a_N) = 4.8$ and $\log(a_N) = 3.3$. This is the error range specified from the experimental benchmarking by Katsura *et al.* The influence of hydrogen activity varying between $\log(a_H) = 1 - 1.5$ had negligible influence and the error bars are convolved with the nitrogen activity (since the relevant molecular specie is ammonia). The revised figure shows that the equilibrium relationships between the two SVON anion-vacancy orderings as a function of strain **are not very sensitive to potential errors in the nitrogen activity**. We have included this new discussion in the revised Supplementary Information. We thank Reviewer #1 for bringing up this important discussion.

Supplementary Figure 9. Activities of nitrogen and oxygen as a function of NH_3 dissociation constant. Our NH_3 flow rate (200 mL/min) is higher than in a previous benchmark (50 mL/min),²² which had a dissociation constant of 0.3. Therefore, our dissociation constant should be lower than 0.3. Below, we include error bars on the free-energy diagram corresponding to a range of nitrogen activities between $\log(a_N) = 4.8$ and $\log(a_N) = 3.3$. Variations in the oxygen and nitrogen stoichiometry between SVON-111 and 112 do not significantly affect the equilibrium relationships between the two anion-vacancy orderings as a function of strain.

REVIEWERS' COMMENTS

Reviewer #1 (Remarks to the Author):

The authors have responded satisfactorily to my previous comments and the manuscript can be accepted for publication.